# Application of Thermo-Malaxation Followed by Three-Phase Centrifugation to Enable the Biorefinery of Alperujo, the Main By-Product of Olive Oil

**DOI:** 10.3390/foods12214023

**Published:** 2023-11-03

**Authors:** África Fernández-Prior, Alejandra Bermúdez-Oria, Fátima Rubio-Senent, Álvaro Villanueva-Lazo, Juan Fernández-Bolaños, Guillermo Rodríguez-Gutiérrez

**Affiliations:** Instituto de la Grasa, Consejo Superior de Investigaciones Científicas (CSIC), Campus Universitario Pablo de Olavide, Edificio 46, Ctra. de Utrera, km 1, 41013 Seville, Spain; mafprior@ig.csic.es (Á.F.-P.); aleberori@ig.csic.es (A.B.-O.); f.r.senent@csic.es (F.R.-S.); alvarovillanueva@ig.csic.es (Á.V.-L.); j.fb.g@csic.es (J.F.-B.)

**Keywords:** olive oil, phenolic compounds, hydroxytyrosol, 3,4-dihydroxyphenylglycol, olive oil solid waste, antioxidants, by-products, thermo-malaxation

## Abstract

The pomace olive oil sector needs to improve its use of the main olive oil by-product, called alperujo, which is currently used mainly for combustion after extraction of pomace oil, with all the problems this process entails due to the high degree of humidity, organic load and phytotoxic substances. In this work, a solution at an industrial level that uses thermo-malaxation at a temperature close to 65 °C for one or two hours followed by centrifugation in three phases is proposed. In this way, over 40% of the pomace oil that is rich in minor compounds, a solid with a lower degree of humidity (55%), and a liquid aqueous fraction that is rich in bioactive compounds such as phenolics and sugars are obtained. This aqueous fraction can be treated through subsequent storage stages to increase its content of the main phenolic, hydroxytyrosol, to up to 1.77 g/L, decreasing its percentage of insoluble solids by up to 1.9%, making it possible to obtain extracts that are rich in hydroxytyrosol using systems that are commonly in place at the industrial level. The aqueous fraction, without phenolics, could be used for energy production. A solid with a slightly higher fat content than the initial alperujo remains, thus the rest of the oil content can be extracted from it using solvent, making it, once defatted, suitable for application in subsequent bioprocesses.

## 1. Introduction

The recent changes taking place in the olive oil industry are mainly aimed at increasing the quality of the final product and improving the management of its by-products. For some time now, technologies have been implemented in the industries that make use of olive oil by-products. These industries, called pomace oil extractors, are the ones that handle the so-called alperujo or main by-product generated from the two-phase olive oil extraction system in Spain. This is the most commonly used system in Spain and other olive oil-producing countries and allows for the generation of a solid with a high percentage of moisture (above 65% on average). Between three and seven million tons of alperujo are generated per year in Spain, which produces almost half of the olive oil consumed worldwide [1]. Although great efforts have been made to minimize its environmental impact, with combustion being the most widely used resource, not all the interesting components in alperujo have been exploited and pollution has not been completely avoided. Beyond its use in simple combustion, its high organic load as well as the presence of compounds with a high phytotoxic capacity have made its proper management difficult for decades [2]. But this disadvantage is actually an advantage as long as its utilization begins with the extraction of these phytotoxins, which are also the most biologically active compounds and are responsible for a large part of the beneficial properties of olive oil [3].

The extraction of one of the main phenolic compounds with the highest activity, hydroxytyrosol [4,5], has already been conducted at an industrial level. A second phenolic compound, 3,4-dihydroxyphenylglycol [5], which is found in lower concentrations but in some cases has greater biological activity than hydroxytyrosol and has shown an important synergistic effect for certain activities such as antioxidant, antiplatelet aggregation and antimicrobial activity, is also being scaled up industrially. In the same way, other phenolic compounds of high interest that can be obtained from alperujo together with another group of bioactive compounds such as carbohydrates, specifically acid and neutral oligosaccharides with or without phenolic remains of a low degree of polymerization, have been described [6]. Obtaining these high-added-value compounds not only increases the value of the by-product but also enables and improves the subsequent application of bioprocesses for obtaining energy (methane and ethanol) [7] or agricultural substrate, as well as direct use in human food (dietary fiber) [8] or animal feed [9]. However, in order to gain access to these bioactive compounds, the pomace must be pre-treated.

The extraction of virgin olive oil is carried out through a continuous system according to the following steps: milling, malaxating below 30 °C and centrifugation in two phases in which two phases are obtained—olive oil and the mixture of the solid fraction and the vegetation water, or also called alperujo. In Spain, it is usual to carry out a second extraction of the alperujo in order to continue extracting oil, although this must be refined. This is done between 35 and 40 °C, obtaining more oil as well as the alperujo. During the milling and the malaxation phases, these compounds are mixed and their binding to the wall material is promoted, making their extraction difficult [10]. This is why it is necessary to apply pre-treatments that can solubilize them in the liquid phase and facilitate subsequent separation by means of filtration or centrifugation. Therefore, a steam-heat treatment system, which operates at a range between 120 and 180 °C, has been developed. A patented system [10], which has been implemented in one of the pomace extractors that treats the largest amount of pomace in Spain, has also been developed. This system allows the main bioactive compounds such as phenols and oligosaccharides to be dissolved into the liquid phase, while fats and proteins are concentrated into the solid phase. In order for the advantages of a pre-treatment to be adopted in the rest of the pomace mills and even in the oil mills, the use of thermo-malaxation at a higher temperature followed by centrifugation was studied in the present work. This system, although not as advantageous as thermal treatment with steam, can allow the initiation of an alperujo biorefinery, making it possible to recover its main bioactive compounds and facilitate the subsequent application of bioprocesses. In this work, the subsequent treatment and storage conditions were also studied in order to adapt the system to the practical requirements necessary for the extraction of bioactive compounds to be developed at an industrial level.

## 2. Materials and Methods

### 2.1. Analytical Methods

#### 2.1.1. Chemicals

The main phenolics like 3,4-dihydroxyphenylglycol or hydroxytyrosol 4-β-d-glucoside were obtained from Sigma-Aldrich (Deisenhofer, Germany), while tyrosol was obtained from Fluka (Buchs, Switzerland) and hydroxytyrosol was obtained from Extrasynthese (Lyon Nord, Geney, France). The eluent for HPLC, namely acetonitrile (HPLC grade) was obtained from Panreac Quimica S.A. (Barcelona, Spain) and ultrapure water was obtained from a Milli-Q water system (Millipore, Milford, MA, USA).

#### 2.1.2. Three Olive Oil Extraction System

Approximately 4000 kg of alperujo samples were collected after two months of storage in ponds and processed in the experimental olive mill at Instituto de la Grasa (CSIC, Sevilla, Spain) where a Pieralisi SPI-7 continuous system was used. This system was designed to produce olive oil with a production of up to 100–120 t/day. It is made up of a 50 HP mill, a paste elevator, a two-body mixer for malaxation (2000 kg capacity each), a mass pump, a vibro-filter, a pump, a vertical centrifuge, an oil receiver tank and an alperujo conveyor screw. The system is controlled using an electric automation panel with a graphic screen to monitor the whole process. First, it was subjected to thermo-malaxation at 65 °C for 90 min and 180 min using Pieralisis equipment (Pieralisis, Jesi, Italy). Then, the alperujo is centrifuged in an adapted three-phase decanter at 3800× *g*, producing a solid phase (SP), an aqueous phase (AP) and pomace olive oil (POO). All samples were stored at −20 °C until analysis.

#### 2.1.3. Phenolic Extraction

The main phenolic compounds present in the alperujo samples and the aqueous fraction after the treatment were extracted following the method described in a previous work [11]. For the alperujo, 20 g of sample were extracted three times with 40 mL of methanol:water (80:20 *v*/*v*). Each extraction was carried out by homogenization using an Ultra Turrax IKA T25 digital blender for 60 s at 1000 rpm. The hydroalcoholic phase was separated by centrifugation for 20 min at 5800× *g* in a Sorvall RT 6000 D centrifuge. The fractions were combined and filtered using a Buchner filter paper, and methanol was evaporated in a vacuum at 45 °C. Finally, the samples were stored at −20 °C until analysis.

#### 2.1.4. Determination of Phenolic Compounds

The main phenolic compounds in the alperujo and the aqueous samples were analyzed using a high-performance liquid chromatography (HPLC) system using a Hewlett-Packard 1100 series equipped with a diode array detector. Quantification was conducted at 280 nm using commercial standards and a comparison with the retention time and UV spectra in the 200–360 nm range was used for identification. The HPLC system has an automatic injector that carries 20 µL of sample. The column used was Teknokroma Tracer Extrasil OSD2 (with a particle size of 5 µm, internal diameter of 250 mm and length of 4.6 mm). A water acidifier with trichloroacetic acid (0.01%) and acetonitrile was used as the mobile phase. The gradient for 55 min was: 95% A initially, 75% A at 30 min, 50% A at 45 min, 0% A at 47 min, 75% A at 50 min, 95% A at 52 min until the end of the run. Quantification was conducted using the regression curve for each phenol. The results were expressed as mg of each phenolic compound per kg of alperujo or per L of aqueous fraction.

#### 2.1.5. Determination of Moisture Content

The moisture content in the alperujo was determined following the indications of the Standard Methods of the American Public Health Association [12]. Approximately 10 g of alperujo sample was dried in a laboratory stove (J.P. Selecta) at 110 °C for 48 h until a constant weight was achieved. The moisture content was expressed as the % of water in fresh alperujo samples.

#### 2.1.6. Determination of Insoluble Solids

Approximately 300 mL of each sample was centrifuged at 15,000× *g* and 4 °C for 30 min in a Sorvall RC-5C refrigerated centrifuge (Du Pont Instruments, Newton, CT, USA). The solid fraction was separated from the liquid and tared rack and left in the oven at 50 °C until a constant weight was achieved. Once dry, the concentration of insoluble solids was expressed as g/L of liquid fraction.

#### 2.1.7. Determination of Fat in the Separated Solid and Liquid

The solid phase was dried at 110 °C and placed in a Soxhlet to extract the fat [11]. The fat content was determined by gravimetric analysis as the difference between the initial and the de-fatted sample in each case.

### 2.2. Methodology for the Aqueous Fraction Post-Treatments

#### 2.2.1. Thermal Treatments

Two temperatures, 70 and 90 °C, were chosen for the treatment of the aqueous fraction (AP) obtained after thermo-malaxation and centrifugation in three phases, for three different periods (30, 60 and 90 min). The temperatures were chosen to be above that previously used (65 °C) and below that of boiling water. Both treatments were carried out by indirect heating in a water bath using a thermostatic bath (PRECISBAT 6 rooms/Tanks 6001482). Agitation was conducted using rotary propeller agitators. The volume of the sample was 500 mL. The use of temperature was also combined with the addition of two concentrations of sulfuric acid (0.5 and 1% *v*/*v* with respect to the volume of water).

#### 2.2.2. Storage of the Aqueous Fraction

Another test that was carried out to increase the HT content and decrease suspended solids involved storing the liquid at room temperature (25 °C) and refrigerating it at 6 °C in a closed tank. Two 100 L tanks and another two refrigerated tanks were kept at room temperature for four months to ensure maximum settling of the fine sediment to favor enzymatic processes and increase the content of simple phenols. Each month, a sample of supernatant was collected to measure the concentration of individual phenolic compounds and insoluble solids.

#### 2.2.3. Vertical Centrifuge

To remove suspended solids from the aqueous fraction (AP), a continuous vertical centrifuge BSGAR250 model (Veronegi, Bologna, Italy) was used.

### 2.3. Statistical Analysis

The results were expressed as mean values with standard deviations. The Statgraphics Plus program, version 2.1, was used to study the differences between samples. Multivariate analysis of variance (ANOVA) followed by Duncan’s comparison test was also performed to compare the differences between groups of samples. The results were considered statistically significant at *p* < 0.05.

## 3. Results and Discussion

### 3.1. Phase Separation

The aim of thermo-malaxation is to promote and facilitate phase separation by solubilizing the compounds with high-added-value biological activity in the liquid phase while improving the utilization of all its phases. Table 1 shows the three separated phases after centrifugation. Starting from an average moisture content in alperujo, treatment at 65 °C for both one and two hours served to reduce the moisture content to 55.1%, with significant differences with respect to time. It also facilitated the extraction of pomace oil, and although the increase over time was not significant, the amount of aqueous phase obtained did increase significantly from one to two hours, owing to the fact that the moisture in the solid phase had also decreased. In this way, the moisture content in the pomace reduced by more than 80%; this could help to reduce the drying costs in the rotary dryers that are normally used in pomace extractors as a step prior to the extraction of pomace oil using a solvent. The drying step is the most energetically costly and problematic in the whole wastewater utilization industry due to the formation of unwanted metabolites such as polycyclic aromatic hydrocarbons [13].

The cost reduction could justify the energy expenditure used in the thermo-malaxation and the subsequent three-stage centrifugation. In addition to this advantage, this system results in higher quality pomace oil where the treatment itself can help to solubilize minor compounds that improve its functionality [14]. This type of pomace oil enriched in minor compounds must be refined. In order to preserve most of these functional compounds, other authors have designed a new physical refining process that would provide oil of higher functional and technological quality [15] and produced functional olive pomace oil. Many studies are being carried out to determine the potential of this functional oil in the prevention of vascular disorders, among other diseases [16,17].

### 3.2. Quantification of Phenolics in the Aqueous Phase

Determination of the phenolic content is key to determining the effectiveness of the process since they are the key to being able to carry out a real biorefinery of the alperujo. These phenolics are those that inhibit the application of subsequent bioprocesses, obtaining energy or agricultural substrate, and above all, they are the components with the greatest added value. The chosen phenolics are the most significant in terms of quantity and biological activity, both in simple and complex forms.

The phenolic analysis performed on the aqueous fractions in the two treatments is shown in Table 2. The increase in total phenols after application of the thermo-malaxation treatment and centrifugation reached 92.6% in the two-hour treatment, with significant differences compared to the one-hour treatment. This increase in phenols is in line with previous reports where the application of higher thermal treatments served to ensure that phenols were not lost during the olive milling and malaxation phases of olive oil production but were bound to the cell wall material in a way that required treatments, such as thermal treatment, to solubilize the phenols and other bioactive compounds [18].

With regard to the main simple phenolics analyzed, HT increased by up to 146.6%. This increase did not correspond to a decrease in the concentration of hydroxytyrosol glycoside, despite this being one of its main precursors [19]. This does not mean that the glycoside was not being hydrolyzed to obtain one molecule of glucose and one molecule of HT, but the amount released by the thermal treatment must equal the amount hydrolyzed.

It is important to bear in mind that the amount of HT–Glu in the olive increases considerably as the fruit ripens, and as the olive pomace used is from the middle of the harvest season, the amount of this phenol should be high [19]. In the case of DHPG, a significant increase was also observed with the treatment and over time, and very high values were obtained in comparison with other heat treatments [20]. It has been observed that treatments above 170 °C cause a degradation of DHPG in the pomace. In this case, the temperature was much lower, hence no such degradation was observed; however, if degradation had occurred, the release and degradation balance would be positive. DHPG has been reported to be a metabolite of hydroxytyrosol [21]. Other precursors of DHPG for example β-hydroxyacteoside, β-hydroxyisoacteoside and 2’´-hydroxyoleuropein, which can increase in concentration in alperujo due to thermal treatments, have been described previously [22].

Tyrosol also shows a significant increase with heat treatment and time. The most notable precursors of tyrosol and HT in olives are secoiridoid phenolics such as ligstroside and oleuropein [23]. These are transformed during the malaxation phase to form more lipophilic derivatives, which are the ones that mainly pass into the oil [24], hence it is these transformed products and other precursors that can further increase the concentration of the phenolic compounds analyzed. This is not the only reason for an increase in these phenolics. It has been shown that in the milling and malaxation phases, phenolic compounds tend to bind to the wall material, more or less complex carbohydrate structures that form cellulose and hemicellulose, thus the application of this type of thermal treatment is necessary to release them [25].

### 3.3. Mass Balance of Thermo-Malaxation and Three-Phase Centrifugation

The balance of the fractions obtained from the application of thermo-malaxation with a retention time of either one or two hours and from the application of three-phase centrifugation is shown in Figure 1. It can be seen that from one ton of pomace, thermo-malaxation gives the best results with two hours of treatment. It reduces the moisture substantially to 55.1%, which has the great advantage, as previously indicated, of reducing drying costs in rotary dryers. This type of dryer was designed not for the drying of the pomace but the solid by-product of the previous three-phase extraction system, which had a much lower moisture content of between 50 and 55%.

The industrial conversion from three-phase to two-phase provided a great advantage for the olive mills but a serious problem for the olive pomace extractors. It was precisely in the rotary dryers where the biggest problems arose, not only because the costs increased substantially, but also because of the technical problems caused by a higher degree of humidity and a higher organic load [26]. This is the reason why the sector has been continuously looking for a solution to this problem. The system studied in this work would return the dryers to the humidity level for which they were designed, reducing operating costs while at the same time solving the problems associated with the formation of undesired substances, which pass into the pomace oil or cause wear and tear of the equipment due to high humidity and caramelization of sugars [27]. On the other hand, this system serves to recover higher quality pomace oil, with a higher content of minor compounds than the pomace olive oil obtained via organic extraction, which could help to revalue this olive pomace oil. Finally, it results in a liquid phase that is enriched in phenolics and other soluble compounds such as sugars and other organic materials, which could be used for agriculture, energy or animal feed [28].

In the present work, water was used to facilitate separation in the three-phase centrifugation system, as a two-phase centrifuge was converted to a three-phase centrifuge by changing the output. Nowadays, centrifuges are evolving in such a way that newer ones that allow three-phase separation do not need water, which implies obtaining a smaller volume of the aqueous liquid fraction that is much more concentrated and therefore more easily usable. This is why this type of centrifuge that does not require the addition of water for three-phase separation would be the most suitable.

The main product obtained from alperujo is pomace oil. Through this system, over 40% of the pomace oil would be obtained physically and with higher quality, which could help revalue this oil by having better functional properties than that obtained with a solvent. To obtain the remaining 60%, it would be necessary to use the hexane extraction system after drying. This system presents many maintenance and environmental problems. The fat content in the starting alperujo was 5.8% fresh matter, which after drying was 16.6%, with a humidity of over 10%. After treatment, the solid fraction, with less moisture, remains at 7.8% fat in fresh matter. This solid fraction must be dried to a humidity of 10%, in which case 17.6% fat content remains. Therefore, despite having extracted a significant amount (more than 40%) of pomace oil via centrifugation, the solid fraction after drying in the rotary dryers would have a greater fat content than the starting pomace due to the dissolving of part of the solid into the aqueous liquid fraction. This is another advantage since the solid has a similar or even higher fat content that would facilitate its use in solvent extraction.

Another important advantage of this system is that as most of the phenolic compounds are solubilized and dissolved into the aqueous liquid fraction, there is a significant reduction in these phenolic compounds in the solid fraction. The phenolic compounds contribute a large portion of the phytotoxicity and inhibit and hinder the application of bioprocesses [29]. This fact has prevented the correct purification of olive oil by-products for several decades. Now, for the first time, the application of this system will make the use of microorganisms more feasible, for example, in the application of aerobic digestion in the production of bioethanol [30], or anaerobic digestion in the production of biomethane [31] or its composting for agricultural use [32], among many other bioprocess options, in order to achieve the true biorefinery of olive pomace.

### 3.4. Aqueous Fraction Post-Treatments

The aqueous liquid fraction obtained after thermo-malaxation and three-phase centrifugation is an important source of bioactive compounds such as sugars and phenolics. To purify phenolic compounds, it is necessary to adapt this aqueous fraction so that the content of the main phenol, hydroxytyrosol, increases and the amount of non-soluble solids decreases to below 3%. These are the requirements necessary for the application of purification systems that are currently in the industry and are fundamentally based on chromatographic systems [33]. Three types of treatments were carried out: the first combined the use of temperature and acid, the second involved sedimentation and the last one was based on centrifugation.

### 3.5. Thermal and Acid Treatments

The results of the treatments at 70 and 90 °C using three concentrations of sulfuric acid and three reaction times are shown in Figure 2. It is clear that the amount of hydroxytyrosol increases with the severity of the treatment, up to values above 1.5 g/L for the two temperatures, which is an increase of over 50% with respect to the starting liquid source (1.01 g/L). In the case of insoluble solids, the behavior is not linear when no acid is used; however, with the use of acid, it decreases to 13 and 8% at 70 and 90 °C, respectively. This gives an indication that part of the solid is dissolved and solubilized and therefore the liquid becomes more clarified at higher temperatures. Longer periods could be used to solubilize the insoluble solid, although the increase in time will mean an increase in cost.

The use of acid improves the release of hydroxytyrosol and the clarification of the liquid, in this case, when carried out at 1%. Not using acid would increase the HT concentration by 25% but would not reduce the percentage of insoluble solids to below 14%. Therefore, to substantially reduce the solid content and increase the HT content, the use of 90 °C and 1% acid would be necessary.

### 3.6. Storage Test

During storage at room temperature, solids settled to a value of 4.9% after four months (Table 3), hence 60% of the total volume was partially clarified while the rest had a higher concentration of solids. In the case of the refrigerated sample, not only was there less settling, reaching only 10% insoluble solids, but these solids increased in the first month due to the decrease in solubilization due to the lower temperature. The simple phenol contents in the supernatants of the two fractions (Table 3) were analyzed.

It is clear that storage at room temperature was very effective and much more so than refrigerated storage where only sedimentation occurred without releasing HT. The amounts of hydroxytyrosol, tyrosol and DHFG increased considerably, while HT glycoside decreased as expected since it was hydrolyzed into its corresponding glucose and hydroxytyrosol molecules.

The increase in HT was 81% compared to the original liquid sample when stored at 25 °C. Under storage, the sedimentation volume, which is discarded and would thus be lost, accounts for 36 and 30% when 25 and 6 °C, respectively, are used. On the other hand, despite the low percentage of solids, the liquid (4.9%) could be further clarified to improve its use in chromatographic purification systems by subjecting the supernatant to centrifugation.

It is important to note that during the four months of storage, the formation of unwanted odors that could influence the production of phenolic extracts was not detected. Traces of 4-ethylphenol, which is a metabolite derived from anaerobic fermentation, were only detected after ten months, a time much longer than that used in the present study.

### 3.7. Centrifugation Test

Due to the need to decrease the insoluble solid content on the one hand and to increase the hydroxytyrosol concentration on the other hand, the aqueous liquid fractions obtained were centrifuged at 65 °C for two hours with and without a subsequent storage phase for four months. A vertical plate centrifuge normally used for cleaning the oil after it leaves the vertical centrifuge was employed. This centrifuge was used due to the fatty nature of the undissolved solids, which are the same as those that are separated from the oil in the oil mill. In the case of the non-stored samples (Table 4), the centrifuge collapsed after a few liters, making it impossible to clarify them by this method; however, the stored liquid sample was centrifuged, reducing its percentage of solids to 1.9%, which improved its subsequent use in the application of chromatographic or membrane systems for the purification of HT.

### 3.8. Final Balance

Figure 3 shows a treatment proposal for improving the management and utilization of alperujo. This scheme involves a first step of thermo-malaxation, followed by three-stage centrifugation from which pomace olive oil and solid and aqueous fractions are produced. A solid with a lower degree of moisture and lower toxicity (because of its lower concentration of phenolic compounds), a crude pomace olive oil that is richer in minor compounds and an aqueous fraction that is rich in bioactive compounds were the final products. To extract phenolic compounds from the aqueous phase, it is necessary to subsequently apply a treatment that on the one hand increases the release of the main phenol, HT, and on the other hand reduces its content in insoluble solids, producing 260 kg of this fraction from one ton of alperujo with a high concentration of HT, which makes its purification possible after storing the fraction for four months and subsequently centrifuging.

## 4. Conclusions

In this work, a new scheme is proposed for the treatment and use of the main by-product in the olive oil industry, alperujo. The application of a work scheme based on thermal-malaxation followed by three-phase centrifugation allows for the production of three more easily usable products. On the one hand, 40% of the pomace oil, which is rich in minor compounds, is obtained by centrifugation, on the other hand, a detoxified solid with much lower humidity (55%), from which the rest of the pomace oil can still be extracted with solvent to be used for energy production, agricultural substrate or animal nutrition, and lastly, an aqueous fraction as a source of bioactive compounds. In order to improve the content of simple phenols in the aqueous fraction and reduce the suspended solids, it is necessary to apply either a heat treatment at 90 °C or prolonged storage followed by centrifugation.

The combined use of thermo-malaxation and three-phase centrifugation is undoubtedly a promising solution that will also make it possible to revalue pomace oil and obtain a new source of bioactive compounds that can be fully exploited through their use as an agricultural substrate, to obtain renewable energy or used as food due to the high amount of sugars. In addition, this scheme takes advantage of elements that are already in the industry itself, proposing a change to the three-phase system in pomace extractors with a prior malaxation step at the highest possible temperature, and connection with another type of industry to achieve the biorefinery of alperujo so that its management is no longer a problem and alperujo becomes a valuable resource. Further studies to achieve full utilization of the treated alperujo, the aqueous phase and the solid detoxified phase to complete the biorefinery will be necessary to assess the industrial condition for bioprocess applications focusing on energy, food or agricultural purposes.

## Figures and Tables

**Figure 1 foods-12-04023-f001:**
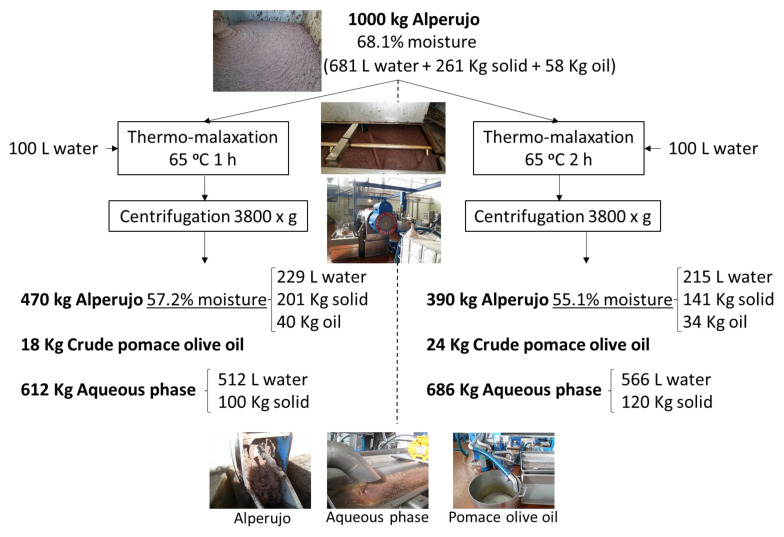
Scheme and mass balance of thermo-malaxation followed by three-phase centrifugation.

**Figure 2 foods-12-04023-f002:**
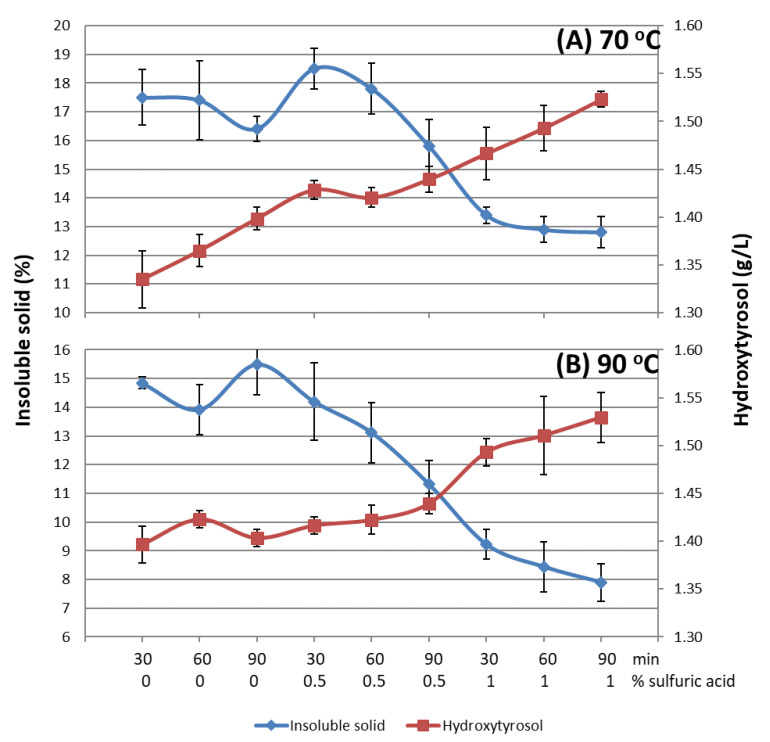
Concentrations of hydroxytyrosol and percentages of insoluble solids in the aqueous fraction obtained after post-thermal treatment at 70 °C (**A**) or 90 °C (**B**), with or without the addition of acid at different times.

**Figure 3 foods-12-04023-f003:**
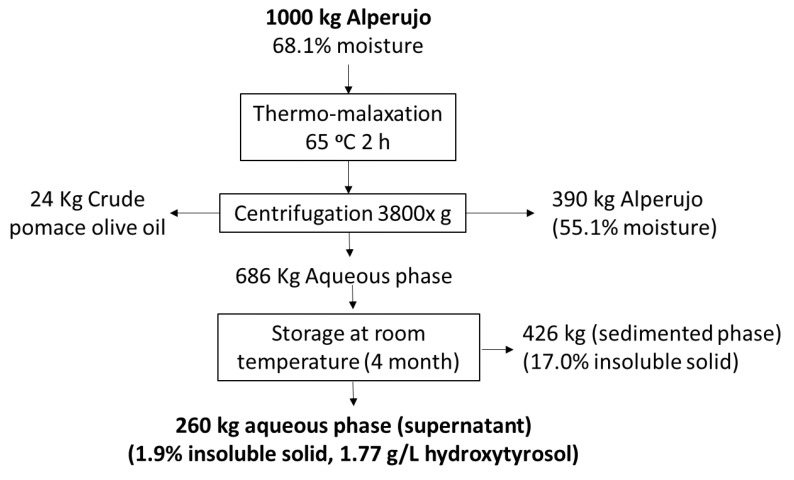
Final balance of the completed treatment from the alperujo to the three different fractions obtained.

**Table 1 foods-12-04023-t001:** Moisture and fat contents in the solid phase before (control) and after thermal treatment and centrifugation, the quantity of crude pomace olive oil and aqueous phase obtained after the thermal treatments and centrifugation and the percentage of insoluble solids in the aqueous phase (*w*/*w*).

Treatment	Solid Phase SP			Pomace Olive Oil POO	Aqueous Phase AP
	Moisture		Fat	g/100 g of Alperujo	g/100 g of Alperujo
	%	% Δ	%		
65 °C, 1 h	57.2 ± 0.4 a *	−16.0	3.4 ± 0.6 a	1.8 ± 0.7	205.4 ± 2.8 b
65 °C, 2 h	55.1 ± 0.8 b	−19.1	3.8 ± 0.2 a	2.4 ± 0.5	253.7 ± 3.3 c
Control	68.1 ± 4.1 b	-	5.8 ± 1.0 b	-	-

* Means with the same letter in the same row were not significantly different. *p* < 0.05.

**Table 2 foods-12-04023-t002:** Total and main individual phenolic compounds in the aqueous phase obtained after the thermal treatments and the centrifugation in comparison with the control. Individual phenolics: 3,4-dihydroxyphenylglycol (DHPG), hydroxytyrosol (HT), Tyrosol (Ty) and hydroxytyrosol 4-β-d-glucoside (Glu-HT).

Treatment	Aqueous Phase (AP)									
	Total Phenolics		Individual Phenolics (mg/L)							
	mg/L		DHPG		HT		Ty		Glu-HT	
	%	% Δ	mg/L	% Δ		% Δ		% Δ		% Δ
65 °C, 1 h	3214.4 ± 24.5 b *	63.3	291.7 ± 8.0 b	49.8	791.5 ± 10.1 b	92.7	205.4 ± 2.8 b	45.9	584.0 ± 12.0 a	−3.8
65 °C, 2 h	3722.1 ± 35.2 c	92.6	335.8 ± 7.9 c	72.5	1012.8 ± 12.2 c	146.6	253.7 ± 3.3 c	80.2	615.3 ± 11.4 a	1.4
Control	1932.8 ± 18.6 a	-	194.7 ± 5.5 a	-	410.7 ± 7.3 a	-	140.8 ± 4.0 a	-	607.1 ± 9.8 a	-

* Means with the same letter in the same row were not significantly different. *p* < 0.05.

**Table 3 foods-12-04023-t003:** Insoluble solids and the main individual phenolics in the supernatant obtained from the stored aqueous phase and the control. Individual phenolic compounds: 3,4-dihydroxyphenylglycol (DHPG), hydroxytyrosol (HT), Tyrosol (Ty) and hydroxytyrosol 4-β-d-glucoside (Glu-HT).

Storage Conditions		Liquid Fraction (Supernatant): Main Simple Phenolics (mg/L)				
Temperature (°C)	Time (Month)	% Insoluble Solids	DHPG	HT	Ty	Glu-HT
25	1	12.4 ± 1.5 cd *	411.2 ± 12.5 c	1102.5 ± 23.5 b	250.4 ± 2 c	420.1 ± 3.8 e
	2	8.5 ± 0.8 b	432.9 ± 14.2 c	1230.4 ± 30.5 c	308.6 ± 3.2 d	280.5 ± 2.1 c
	3	7.0 ± 0.5 b	458.1 ± 13.0 c	1594.7 ± 18.9 d	355.0 ± 1.9 e	163.9 ± 1.9 b
	**4**	**4.9 ± 0.4 a**	**523.0 ± 9.8 d**	**1770.5 ± 20.1 e**	**430.3 ± 3.5 f**	**60.5 ± 0.6 a**
6	1	14.7 ± 0.9 d	364.4 ± 10.7 b	978.3 ± 12.7 a	240.2 ± 2.1 c	352.6 ± 4.4 d
	2	12.6 ± 0.8 c	372.5 ± 10.0 b	980.5 ± 10.6 a	232.0 ± 1.1 c	310.4 ± 3.8 cd
	3	11.2 ± 0.7 bc	355.3 ± 8.2 ab	986.1 ± 11.8 a	200.7 ± 2.8 b	296.7 ± 2.0 c
	**4**	**10.0 ± 0.3 b**	**330.1 ± 7.6 a**	**990.8 ± 13.4 a**	**163.5 ± 1.5 a**	**236.5 ± 1.8**
Control	-	18.7 ± 1.2 e	387.1 ± 9.1 b	980.3 ± 12.5 a	234.5 ± 2.9 c	511.0 ± 4.5 f

* Means with the same letter in the same row were not significantly different. *p* < 0.05.

**Table 4 foods-12-04023-t004:** Results of centrifugation tests of the non-stored aqueous fraction treated at 65 °C for 2 h and the fraction stored for four months at room temperature.

Sample Treated at 65 °C, 2 h	% Initial Insoluble Solid	Centrifuged Volume (L)	Supernatant Fraction		Precipitated Fraction	
			% Final Insoluble Solid	Volume (%)	% Insoluble Solid	Volume (%)
No stored	18.2 ± 2.0	5	-	32 ± 1	18.2 ± 2.0	100
Stored for 4 months at 25 °C	4.9 ± 0.4	50	1.9 ± 0.2	38 ± 2	17.0 ± 0.8	62 ± 3

## Data Availability

The data are contained within the article.

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
