# Peer review of "Application of Thermo-Malaxation Followed by Three-Phase Centrifugation to Enable the Biorefinery of Alperujo, the Main By-Product of Olive Oil"

_foods, 2023, doi:10.3390/foods12214023_

Round 1

Reviewer 1 Report

Comments and Suggestions for Authors

The valorisation of the by-products of the mechanical extraction of virgin olive oils is a topic of great interest for improving the productivity and sustainability of the olive oil supply chain. The proposed work is therefore interesting as a topic but, from a scientific point of view, shows some limitations. The first point relates to the evaluation of the phenolic compounds which were carried out only on the liquid phase of the process while, to define a correct mass balance, the phenolic concentration of the pomace should also have been analyzed after malaxation at high temperature and separation by centrifugation of the liquid phase (vegetation waters). Another important point is that the pomace not only contains hydroxytyrosol, tyrosol and hydroxytyrosol glucoside but also other secoiridoids such as 3,4-DHPEA-EDA, p-HPEA-EDA, eleuropein aglycone and ligustroside aglycone which, upon hydrolysis, they could produce hydroxytyrosol and tyrosol during both the high-temperature malaxation process, the acid hydrolysis phase and the vegetation waters storage phase. Therefore, a complete characterization of the phenolic composition of the samples should be reported. An additional aspect concerns the determination of the total phenolic compounds of the pomace and vegetation waters produced. The reported data only show the transfer of phenolics from the pomace to the vegetation waters, after the high temperature malaxation treatment, but do not show the level of oxidation of the phenolic compounds during the process. A final point concerns the production of off-flavours during the storage of vegetation waters at room temperature and 4 °C. The volatile compounds produced by fermentation of the vegetation waters can significantly complicate the recovery and purification processes of phenolic compounds from vegetation waters, for this reason a study of microbial development and production of volatile compounds during the storage of vegetation waters should be reported.

Comments on the Quality of English Language

No comments

Author Response

Reviewer 1

The valorisation of the by-products of the mechanical extraction of virgin olive oils is a topic of great interest for improving the productivity and sustainability of the olive oil supply chain. The proposed work is therefore interesting as a topic but, from a scientific point of view, shows some limitations. The first point relates to the evaluation of the phenolic compounds which were carried out only on the liquid phase of the process while, to define a correct mass balance, the phenolic concentration of the pomace should also have been analyzed after malaxation at high temperature and separation by centrifugation of the liquid phase (vegetation waters).

Response: Phenolic analyses were performed before and after thermo-malaxation and centrifugation. As specified in materials and methods, after the thermo-malaxation a centrifugation is applied, without which it would not be possible to separate the liquid and solid phases, and it is precisely on the liquid phase that the phenol analysis was carried out. To avoid confusion, in tables one and two it has been specified that the liquid fraction analysed was the one obtained after applying not only the thermal treatment or thermo-malaxation but also the centrifugation.

 Another important point is that the pomace not only contains hydroxytyrosol, tyrosol and hydroxytyrosol glucoside but also other secoiridoids such as 3,4-DHPEA-EDA, p-HPEA-EDA, eleuropein aglycone and ligustroside aglycone which, upon hydrolysis, they could produce hydroxytyrosol and tyrosol during both the high-temperature malaxation process, the acid hydrolysis phase and the vegetation waters storage phase. Therefore, a complete characterization of the phenolic composition of the samples should be reported.

Response: As the reviewer rightly says, pomace contains not only the phenolics analyzed, but also derivatives of secoiridoids, although after olive oil extraction most of these, such as oleacein and oleocanthal, pass preferentially to the oily phase, leaving very little in the alperujo or pomace. However, it is precisely the phenolics analyzed that form the basis of most of them and are present in the pomace in the greatest quantity, in the case of hydroxytyrosol they can reach almost 50 % of the total phenolics in the pomace, and in the case of vegetation waters they are undoubtedly the majority in both free or conjugate form. It is for this reason that these phenolics have been chosen, as they show the behaviour of the treatments applied on the alperujo.

An additional aspect concerns the determination of the total phenolic compounds of the pomace and vegetation waters produced. The reported data only show the transfer of phenolics from the pomace to the vegetation waters, after the high temperature malaxation treatment, but do not show the level of oxidation of the phenolic compounds during the process.

Response: Thermal treatments, such as steam explosion treatment, or hydrothermal treatment, served to scientifically prove that the phenols in the pomace matrix were not oxidized in the milling or malaxation phase, while the high resistance to oxidation of these phenols during the application of high temperatures, pressures and time in the pomace was verified. Thanks to this, it was possible to initiate the recovery of these phenolic at industrial scale after thermal treatments. The temperature range in which the HT, Ty and DHPG precursors are hydrolyzed was also verified. That is why these phenols have been used as markers of the hydrolysis reactions, obviating the formation of oxidation metabolites which are not formed at such low temperature, as has already been demonstrated. References:

  • Fernández-Bolaños, et al. “Production in large quantities of highly purified hydroxytyrosol from liquid-solid waste of two-phase olive oil processing or Alperujo”. J. Agric. Food Chem. (2002) 50: 6804-6811. https://doi.org/10.1021/jf011712r
  • Fernández-Bolaños, et al. “Total Recovery of the Waste of Two-Phase Olive oil Processing: Insolation of Added-Value Compounds”. J. Agric. Food Chem. (2004) 52: 5849-5855. https://doi.org/10.1021/jf030821y
  • Fernández-Bolaños, et al. “Production in large quantities of highly purified hydroxytyrosol from liquid-solid waste of two-phase olive oil processing or Alperujo”. J. Agric. Food Chem. (2002) 50: 6804-6811. https://doi.org/10.1021/jf011712r
  • Fernández-Bolaños, et al. “Total Recovery of the Waste of Two-Phase Olive oil Processing: Insolation of Added-Value Compounds”. J. Agric. Food Chem. (2004) 52: 5849-5855. https://doi.org/10.1021/jf030821y
  • Lama-Muñoz, A., et al. A study of the precursors of the natural antioxidant phenol 3,4-dihydroxyphenylglycol in olive oil waste. Food Chemistry (2013), 140, 154-160. https://doi.org/10.1016/j.foodchem.2013.02.063

 A final point concerns the production of off-flavours during the storage of vegetation waters at room temperature and 4 °C. The volatile compounds produced by fermentation of the vegetation waters can significantly complicate the recovery and purification processes of phenolic compounds from vegetation waters, for this reason a study of microbial development and production of volatile compounds during the storage of vegetation waters should be reported.

Response: Very good appreciation from the reviewer. Without a doubt, it is a very striking aspect, since the liquid fraction is enriched in phenolics, with a fairly high total concentration that allows maintaining a very low microbiological level and thus also prevents the appearance of unwanted odors. Only the formation of some yeast colonies when storage time is longer than three months. These liquid fractions are being used at an industrial level and the companies in charge of this carry out continuous microbial monitoring, verifying the absence of pathogenic microorganisms. These data could not be shown as they are part of these companies. However, a paragraph has been unwanted that details the absence of unwanted odors and the fact that only after ten months has it been possible to identify one of the markers of these unwanted reactions, such as 4-ethylphenol.

Reviewer 2 Report

Comments and Suggestions for Authors

In this manuscript, thermo-malaxation of alperujo followed by three-phase centrifugation was investigated. It is very interesting subject which can solve a important problem and issue in the olive oil mills. But, the presentation and writing of the manuscript is confusing and complex. Abstract should have scientific structure. One or two introduction sentences and then methodology and results. But in this manuscript, it was not complied. Also, there should be some data in the abstract to show the clear picture of the findings.

Method and materials part also has not in the journal format. First, chemicals should be written and the the methods. Figure one and flowchart of the work up should be in methodology part even though without data.

Different subjects are mixed up in the methodology, it could be better to divide the research steps and method part as well to better and fast understanding of the work.

There are some weak points in the results section as well. The format of the results and discussion section in a scientific report is as follow, first presenting some justification to show why the experiment has been done, then presenting the obtained results and then discussion and comparing with the already published data. This has not been done and observed in this manuscript. Many interesting results but sorry to say with weak explanation and presentation.

Table titles should be short, more explanations should be in Table footnotes.Is it economical to obtain 1.5 or 2.4 % of oil as it is presented in Table 1?

It is suggested to revise this manuscript extensively specially in the writing and presentation of the work. There are many complex paragraphs, foe example, lines 208-213, or 218 to 222. Actually, this is for whole manuscript. It needs very serious revision and rewriting and clear presentation.

Comments on the Quality of English Language

It needs very serious revision and rewriting and clear presentation.

Author Response

Reviewer 2

In this manuscript, thermo-malaxation of alperujo followed by three-phase centrifugation was investigated. It is very interesting subject which can solve an important problem and issue in the olive oil mills. But, the presentation and writing of the manuscript is confusing and complex. Abstract should have scientific structure. One or two introduction sentences and then methodology and results. But in this manuscript, it was not complied. Also, there should be some data in the abstract to show the clear picture of the findings.

Response: Due to the limitation of words, the abstract has been simplified as much as possible, but it, has been written following the scientific structure:

Introduction: The pomace olive oil sector needs to improve its use of the main olive oil by-product, called alperujo, which is currently used mainly for combustion after the extraction of the pomace oil, with all the problems this process entails, due to the high degree of humidity, organic load and phytotoxic substances.

Material and method: In this work, a solution is proposed at an industrial level with the use of thermo-malaxation at a temperature close to 65 ºC for one or two hours, followed by centrifugation in three phases.

Results (with more remarkable dates): In this way, over 40% of the pomace oil, which is rich in minor compounds, a solid with a lower degree of humidity (55%), and a liquid aqueous fraction which is rich in bi-oactive compounds, such as phenolics and sugars are obtained. This aqueous fraction can be treated through subsequent storage stages to increase its content in the main phenolic, hydroxy-tyrosol, to up to 1.77 g/L, decreasing its percentage of insoluble solids by up to 1.9%, making it possible to obtain extracts which are rich in hydroxytyrosol using the systems commonly in place at the industrial level. The aqueous fraction without phenolics could be used for energy production. A solid with a slightly higher fat content than the initial alperujo remains, so the rest of the oil content can be extracted from it by solvent, making it, once defatted, suitable for the application of subsequent bioprocesses.

Method and materials part also has not in the journal format. First, chemicals should be written and the methods.

Response: The chemicals section has been put first.

Figure one and flowchart of the work up should be in methodology part even though without data.

Response: The work scheme is very simple, that is why no figure has been included in this section and it has been preferred to detail a final balance in results.

Different subjects are mixed up in the methodology, it could be better to divide the research steps and method part as well to better and fast understanding of the work.

Response: To clarify this section,  three point has been used:  analytical analysis, methodology and statistical analysis.

There are some weak points in the results section as well. The format of the results and discussion section in a scientific report is as follow, first presenting some justification to show why the experiment has been done, then presenting the obtained results and then discussion and comparing with the already published data. This has not been done and observed in this manuscript. Many interesting results but sorry to say with weak explanation and presentation.

Response: It must be understood that this work is the result of many years of research and represents an important compendium that will help improve the management of the main by-product of olive oil. In this work, a methodology very focused on its industrial scale is used, supported by analytical determinations, which is why it is not a basic research work but an applied one. The results have been presented in a scientific manner, the only thing missing was an introductory paragraph in the analysis of phenolics that has already been inserted in the text. In the rest of the results section, the objective and results are clearly explained and discussed in comparison with other authors or processes.

Table titles should be short, more explanations should be in Table footnotes. Is it economical to obtain 1.5 or 2.4 % of oil as it is presented in Table 1?

Response: Most of the explanation is about the full name of the phenols that must accompany each table to avoid confusion. By other hand, the industrial quantities of alperujo handled are very high, more than five million tons per year. The extraction that is being carried out in the industry is mostly chemical, with n-hexane, which makes pomace oil very expensive. In this proposal it is intended that the majority of the oil be obtained by centrifugation, that is to say physically, but also with a higher content of minority components, which will help the sector to produce a higher quality pomace oil, even if it is in least amount.

It is suggested to revise this manuscript extensively specially in the writing and presentation of the work. There are many complex paragraphs, foe example, lines 208-213, or 218 to 222. Actually, this is for whole manuscript. It needs very serious revision and rewriting and clear presentation.

Response: The complexity of the results corresponds to the complexity of being able to describe the changes and reactions that occur with phenolics. It has been described in the simplest and most direct way possible, although it is true to have knowledge about the phenols of olives to be able to understand it well. Describing the behavior of phenols is very complicated, but in this work it has been presented in the simplest way possible. It must also be taken into account that it is a research work with a high degree of application carried out with laboratory and pilot plant tests focused on its industrial scale, which is why it moves away from a purely basic research work.

Round 2

Reviewer 1 Report

Comments and Suggestions for Authors

The valorisation of the by-products of the mechanical extraction of virgin olive oils is a topic of great interest for improving the productivity and sustainability of the olive oil supply chain. The proposed work is therefore interesting as a topic but, from a scientific point of view, shows some limitations. The first point relates to the evaluation of the phenolic compounds which were carried out only on the liquid phase of the process while, to define a correct mass balance, the phenolic concentration of the pomace should also have been analysed after malaxation at high temperature and separation by centrifugation of the liquid phase (vegetation waters).

Response: Phenolic analyses were performed before and after thermo-malaxation and centrifugation. As specified in materials and methods, after the thermo-malaxation a centrifugation is applied, without which it would not be possible to separate the liquid and solid phases, and it is precisely on the liquid phase that the phenol analysis was carried out. To avoid confusion, in tables one and two it has been specified that the liquid fraction analysed was the one obtained after applying not only the thermal treatment or thermo-malaxation but also the centrifugation.

Referee repose

The authors report in the materials and methods that a three-phase decanter was used for the separation of the liquid phase after the thermal treatment. the Decanter is used to carry out a solid-liquid separation so it seems really strange that from the horizontal centrifuge two phases have not been separated: the aqueous phase and a second phase which is actually solid even if with high humidity. Reading the tables as they were reported between control and treated at high temperature it seems that the heat treatment process generates phenolic compounds instead of producing their transfer from the solid phase of the pomace to the liquid phase

Another important point is that the pomace not only contains hydroxytyrosol, tyrosol and hydroxytyrosol glucoside but also other secoiridoids such as 3,4-DHPEA-EDA, p-HPEA-EDA, eleuropein aglycone and ligustroside aglycone which, upon hydrolysis, they could produce hydroxytyrosol and tyrosol during both the high-temperature malaxation process, the acid hydrolysis phase and the vegetation waters storage phase. Therefore, a complete characterization of the phenolic composition of the samples should be reported.

Response: As the reviewer rightly says, pomace contains not only the phenolics analyzed, but also derivatives of secoiridoids, although after olive oil extraction most of these, such as oleacein and oleocanthal, pass preferentially to the oily phase, leaving very little in the alperujo or pomace. However, it is precisely the phenolics analyzed that form the basis of most of them and are present in the pomace in the greatest quantity, in the case of hydroxytyrosol they can reach almost 50 % of the total phenolics in the pomace, and in the case of vegetation waters they are undoubtedly the majority in both free or conjugate form. It is for this reason that these phenolics have been chosen, as they show the behaviour of the treatments applied on the alperujo.

Referee repose

The fresh pomaces just separated from the decanter contains a high concentration of aglycon derivative of secoriridoids, such as oleocanthal oleacin, and in several cases, verbascoside which is cultivar dependent. As consequence, it is not correct to affirm that the aglycone derivatives of the secoiridoids are more soluble in oil than in the liquid phase. In fact, by carrying out a mass balance, approximately 2-3% of the phenolic compounds of the olive pastes are released into the oil after pressing, the others are they distribute between vegetation water and virgin pomace. The authors should instead point out that the pomace used in their experimental work was probably pomace subjected to a prolonged period of storage. In this case due to the acid hydrolysis of the aglycone derivatives of secoiridoids which is promoted by the acidification of the pomace due to the fermentation processes, a strong reduction of oleacin and oleocanthal should be observed with the corresponding increase in free hydroxytyrosol and tyrosol.

In conclusion the work should be reviewed more extensively 

Author Response

The valorisation of the by-products of the mechanical extraction of virgin olive oils is a topic of great interest for improving the productivity and sustainability of the olive oil supply chain. The proposed work is therefore interesting as a topic but, from a scientific point of view, shows some limitations. The first point relates to the evaluation of the phenolic compounds which were carried out only on the liquid phase of the process while, to define a correct mass balance, the phenolic concentration of the pomace should also have been analysed after malaxation at high temperature and separation by centrifugation of the liquid phase (vegetation waters).

Response 1: Phenolic analyses were performed before and after thermo-malaxation and centrifugation. As specified in materials and methods, after the thermo-malaxation a centrifugation is applied, without which it would not be possible to separate the liquid and solid phases, and it is precisely on the liquid phase that the phenol analysis was carried out. To avoid confusion, in tables one and two it has been specified that the liquid fraction analysed was the one obtained after applying not only the thermal treatment or thermo-malaxation but also the centrifugation.

Referee repose

The authors report in the materials and methods that a three-phase decanter was used for the separation of the liquid phase after the thermal treatment. the Decanter is used to carry out a solid-liquid separation so it seems really strange that from the horizontal centrifuge two phases have not been separated: the aqueous phase and a second phase which is actually solid even if with high humidity. Reading the tables as they were reported between control and treated at high temperature it seems that the heat treatment process generates phenolic compounds instead of producing their transfer from the solid phase of the pomace to the liquid phase

Response 2: The two-phase horizontal centrifuge extraction system is designed to separate only two phases: the oil and the mixture of the solid and aqueous liquid fractions. In other words, it separates the oil from the rest (solid plus vegetation water) on one side. However, it is the three-phase vertical centrifuge extraction system that does separate three phases: an oil fraction and a solid fraction with a moisture content of about 55 %, and an aqueous liquid fraction (vegetation water plus any water that may be added during malaxation).  This principle is the basis of the article, and to make it clearer the following paragraph has been inserted in the introduction: “The extraction of virgin olive oil is carried out through a continuous system by the following steps, milling, malaxating below 30 ºC and centrifugation in two phases in which two phases are obtained olive oil and the mixture of the solid fraction and the vegetation water, or also called alperujo. In Spain, it is usual to make a second extraction of the alperujo in order to continue extracting oil, although this must be refined. This is done between 35 and 40 ºC, obtaining more oil and again the alperujo".

Therefore, the aqueous phase is not separated from the solid phase in the oil mills and this is what is called alperujo. In this work it is proposed to use a three-phase system, not in the olive oil milling but in the olive pomace extractors, that allows the separation of the aqueous and solid phases, obtaining in addition a pomace oil by centrifugation. Increasing the temperature in the malaxation allows the phenols to be solubilized from the solid phase to the aqueous phase, as many of our previous works cited in this article have shown. Phenols were shown to bind to the cell wall material and treatments such as heat treatment had to be used to separate them solubilizing to the aqueous fraction.

Another important point is that the pomace not only contains hydroxytyrosol, tyrosol and hydroxytyrosol glucoside but also other secoiridoids such as 3,4-DHPEA-EDA, p-HPEA-EDA, eleuropein aglycone and ligustroside aglycone which, upon hydrolysis, they could produce hydroxytyrosol and tyrosol during both the high-temperature malaxation process, the acid hydrolysis phase and the vegetation waters storage phase. Therefore, a complete characterization of the phenolic composition of the samples should be reported.

Response 1: As the reviewer rightly says, pomace contains not only the phenolics analyzed, but also derivatives of secoiridoids, although after olive oil extraction most of these, such as oleacein and oleocanthal, pass preferentially to the oily phase, leaving very little in the alperujo or pomace. However, it is precisely the phenolics analyzed that form the basis of most of them and are present in the pomace in the greatest quantity, in the case of hydroxytyrosol they can reach almost 50 % of the total phenolics in the pomace, and in the case of vegetation waters they are undoubtedly the majority in both free or conjugate form. It is for this reason that these phenolics have been chosen, as they show the behaviour of the treatments applied on the alperujo.

Referee repose

The fresh pomaces just separated from the decanter contains a high concentration of aglycon derivative of secoriridoids, such as oleocanthal oleacin, and in several cases, verbascoside which is cultivar dependent. As consequence, it is not correct to affirm that the aglycone derivatives of the secoiridoids are more soluble in oil than in the liquid phase. In fact, by carrying out a mass balance, approximately 2-3% of the phenolic compounds of the olive pastes are released into the oil after pressing, the others are they distribute between vegetation water and virgin pomace. The authors should instead point out that the pomace used in their experimental work was probably pomace subjected to a prolonged period of storage. In this case due to the acid hydrolysis of the aglycone derivatives of secoiridoids which is promoted by the acidification of the pomace due to the fermentation processes, a strong reduction of oleacin and oleocanthal should be observed with the corresponding increase in free hydroxytyrosol and tyrosol.

Response 2: We fully agree with the reviewer, as the alperujo we have used in this work has been stored for at least two months in the ponds. This is why this point has been detailed in the article: “Alperujo samples were collected after two months of storage in the ponds”.

Reviewer 2 Report

Comments and Suggestions for Authors

Majority of the comments and suggestion were included in the manuscript and its acceptance is suggested.

Author Response

Thanks to the reviewer for his contributions which have helped to improve the quality of the article.